# Gradient-based Sampling: An Adaptive Importance Sampling for Least-squares

**Rong Zhu**

Academy of Mathematics and Systems Science, Chinese Academy of Sciences, Beijing, China.
rongzhu@amss.ac.cn

## Abstract

In modern data analysis, random sampling is an efficient and widely-used strategy to overcome the computational difficulties brought by large sample size. In previous studies, researchers conducted random sampling which is according to the input data but independent on the response variable, however the response variable may also be informative for sampling. In this paper we propose an adaptive sampling called the *gradient-based sampling* which is dependent on both the input data and the output for fast solving of least-square (LS) problems. We draw the data points by random sampling from the full data according to their gradient values. This sampling is computationally saving, since the running time of computing the sampling probabilities is reduced to $O(nd)$ where $n$ is the full sample size and $d$ is the dimension of the input. Theoretically, we establish an error bound analysis of the general importance sampling with respect to LS solution from full data. The result establishes an improved performance of the use of our *gradient-based sampling*. Synthetic and real data sets are used to empirically argue that the *gradient-based sampling* has an obvious advantage over existing sampling methods from two aspects of statistical efficiency and computational saving.

## 1 Introduction

Modern data analysis always addresses enormous data sets in recent years. Facing the increasing large sample data, computational savings play a major role in the data analysis. One simple way to reduce the computational cost is to perform random sampling, that is, one uses a small proportion of the data as a surrogate of the full sample for model fitting and statistical inference. Among random sampling strategies, uniform sampling is simple but trivial way since it fails to exploit the unequal importance of the data points. As an alternative, leverage-based sampling is to perform random sampling with respect to nonuniform sampling probabilities that depend on the empirical statistical leverage scores of the input matrix $\mathbf{X}$. It has been intensively studied in the machine learning community and has been proved to achieve much better results for worst-case input than uniform sampling [1, 2, 3, 4]. However it is known that leverage-based sampling replies on input data but is independent on the output variable, so does not make use of the information of the output. Another shortcoming is that it needs to cost much time to get the leverage scores, although approximating leverage scores has been proposed to further reduce the computational cost [5, 6, 7].

In this paper, we proposed an adaptive importance sampling, the *gradient-based sampling*, for solving least-square (LS) problem. This sampling attempts to sufficiently make use of the data information including the input data and the output variable. This adaptive process can be summarized as follows: given a pilot estimate (good "guess") for the LS solution, determine the importance of each data point by calculating the gradient value, then sample from the full data by importance sampling according to the gradient value. One key contribution of this sampling is to save more computational time than leverage-based sampling, and the running time of getting the probabilities is reduced to

$O(nd)$ where $n$ is the sample size and $d$ is the input dimension. It is worthy noting that, although we apply *gradient-based sampling* into the LS problem, we believe that it may be extended to fast solve other large-scale optimization problems as long as the gradient of optimization function is obtained. However this is out of the scope so we do not extend it in this paper.

Theoretically, we give the risk analysis, error bound of the LS solution from random sampling. [8] and [9] gave the risk analysis of approximating LS by Hadamard-based projection and covariance-thresholded regression, respectively. However, no such analysis is studied for importance sampling. The error bound analysis is a general result on any importance sampling as long as the conditions hold. By this result, we establishes an improved performance guarantee on the use of our *gradient-based sampling*. It is improved in the sense that our *gradient-based sampling* can make the bound approximately attain its minimum, while previous sampling methods can not get this aim. Additionally, the non-asymptotic result also provides a way of balancing the tradeoff between the subsample size and the statistical accuracy.

Empirically, we conduct detailed experiments on datasets generated from the mixture Gaussian and real datasets. We argue by these empirical studies that the *gradient-based sampling* is not only more statistically efficient than leverage-based sampling but also much computationally cheaper from the computational viewpoint. Another important aim of detailed experiments on synthetic datasets is to guide the use of the sampling in different situations that users may encounter in practice.

The remainder of the paper is organized as follows: In Section 2, we formally describe random sampling algorithm to solve LS, then establish the *gradient-based sampling* in Section 3. The non-asymptotic analysis is provided in Section 4. We study the empirical performance on synthetic and real world datasets in Section 5.

*Notation*: For a symmetric matrix $\mathbf{M} \in \mathbb{R}^{d \times d}$, we define $\lambda_{\min}(\mathbf{M})$ and $\lambda_{\max}(\mathbf{M})$ as its the largest and smallest eigenvalues. For a vector $\boldsymbol{v} \in \mathbb{R}^d$, we define $\|\boldsymbol{v}\|$ as its $L^2$ norm.

## 2   Problem Set-up

For LS problem, suppose that there are an $n \times d$ matrix $\mathbf{X} = (\boldsymbol{x}_1, \cdots, \boldsymbol{x}_n)^T$ and an $n \times 1$ response vector $\boldsymbol{y} = (y_1, \cdots, y_n)^T$. We focus on the setting $n \gg d$. The LS problem is to minimize the sample risk function of parameters $\boldsymbol{\beta}$ as follows:

$$\sum_{i=1}^n (y_i - \boldsymbol{x}_i^T \boldsymbol{\beta})^2 / 2 =: \sum_{i=1}^n l_i. \tag{1}$$

The solution of equation (1) takes the form of

$$\hat{\boldsymbol{\beta}}_n = (n^{-1} \mathbf{X}^T \mathbf{X})^{-1} (n^{-1} \mathbf{X}^T \boldsymbol{y}) =: \boldsymbol{\Sigma}_n^{-1} \boldsymbol{b}_n, \tag{2}$$

where $\boldsymbol{\Sigma}_n = n^{-1} \mathbf{X}^T \mathbf{X}$ and $\boldsymbol{b}_n = n^{-1} \mathbf{X}^T \boldsymbol{y}$. However, the challenge of large sample size also exists in this simple problem, i.e., the sample size $n$ is so large that the computational cost for calculating LS solution (2) is very expensive or even not affordable.

We perform the random sampling algorithm as follows:
(a) Assign sampling probabilities $\{\pi_i\}_{i=1}^n$ for all data points such that $\sum_{i=1}^n \pi_i = 1$;
(b) Get a subsample $S = \{(\boldsymbol{x}_i, y_i) : i \text{ is drawn}\}$ by random sampling according to the probabilities;
(c) Maximize a weighted loss function to get an estimate $\tilde{\boldsymbol{\beta}}$

$$\tilde{\boldsymbol{\beta}} = \arg \min_{\boldsymbol{\beta} \in \mathbb{R}^d} \sum_{i \in S} \frac{1}{2\pi_i} \|y_i - \boldsymbol{x}_i^T \boldsymbol{\beta}\|^2 = \boldsymbol{\Sigma}_s^{-1} \boldsymbol{b}_s, \tag{3}$$

where $\boldsymbol{\Sigma}_s = \frac{1}{n} \mathbf{X}_s^T \boldsymbol{\Phi}_s^{-1} \mathbf{X}_s$, $\boldsymbol{b}_s = \frac{1}{n} \mathbf{X}_s^T \boldsymbol{\Phi}_s^{-1} \boldsymbol{y}_s$, and $\mathbf{X}_s$, $\boldsymbol{y}_s$ and $\boldsymbol{\Phi}_s$ are the partitions of $\mathbf{X}$, $\boldsymbol{y}$ and $\boldsymbol{\Phi} = \text{diag}\{r\pi_i\}_{i=1}^n$ with the subsample size $r$, respectively, corresponding the subsample $S$. Note that the last equality in (3) holds under the assumption that $\boldsymbol{\Sigma}_s$ is invertible. Throughout this paper, we assume that $\boldsymbol{\Sigma}_s$ is invertible for the convenience since $p \ll n$ in our setting and it can be replaced with its regularized version if it is not invertible.

How to construct $\{\pi_i\}_{i=1}^n$ is a key component in random sampling algorithm. One simple method is the uniform sampling, i.e.,$\pi_i = n^{-1}$, and another method is leverage-based sampling, i.e., $\pi_i \propto$

$\boldsymbol{x}_i^T(\mathbf{X}^T\mathbf{X})^{-1}\boldsymbol{x}_i$. In the next section, we introduce a new efficient method: *gradient-based sampling*, which draws data points according to the gradient value of each data point.

**Related Work.** [10, 11, 4] developed leverage-based sampling in matrix decomposition. [10, 12] applied the sampling method to approximate the LS solution. [13] derived the bias and variance formulas for the leverage-based sampling algorithm in linear regression using the Taylor series expansion. [14] further provided upper bounds for the mean-squared error and the worst-case error of randomized sketching for the LS problem. [15] proposed a sampling-dependent error bound then implied a better sampling distribution by this bound. Fast algorithms for approximating leverage scores $\{\boldsymbol{x}_i^T(\mathbf{X}^T\mathbf{X})^{-1}\boldsymbol{x}_i\}_{i=1}^n$ were proposed to further reduce the computational cost [5, 6, 7].

## 3 Gradient-based Sampling Algorithm

The *gradient-based sampling* uses a pilot solution of the LS problem to compute the gradient of the objective function, and then sampling a subsample data set according to the calculated gradient values. It differs from leverage-based sampling in that the sampling probability $\pi_i$ is allowed to depend on input data $\mathbf{X}$ as well as $\boldsymbol{y}$. Given a pilot estimate (good guess) $\boldsymbol{\beta}_0$ for parameters $\boldsymbol{\beta}$, we calculate the gradient for the $i$th data point

$$\boldsymbol{g}_i = \frac{\partial l_i(\boldsymbol{\beta}_0)}{\partial \boldsymbol{\beta}_0} = \boldsymbol{x}_i(y_i - \boldsymbol{x}_i^T\boldsymbol{\beta}_0). \tag{4}$$

Gradient represents the slope of the tangent of the loss function, so logically if gradient of data points are large in some sense, these data points are important to find the optima. Our sampling strategy makes use of the gradient upon observing $y_i$ given $\boldsymbol{x}_i$, and specifically,

$$\pi_i^0 = \|\boldsymbol{g}_i\| / \sum_{i=1}^n \|\boldsymbol{g}_i\|. \tag{5}$$

Equations (4) and (5) mean that, $\|\boldsymbol{g}_i\|$ includes two parts of information: one is $\|\boldsymbol{x}_i\|$ which is the information provided by the input data and the other is $|y_i - \boldsymbol{x}_i^T\boldsymbol{\beta}_0|$ which is considered to provide a justification from the pilot estimate $\boldsymbol{\beta}_0$ to a better estimate. Figure 1 illustrates the efficiency benefit of the *gradient-based sampling* by constructing the following simple example. The figure shows that the data points with larger $|y_i - x_i\beta_0|$ are probably considered to be more important in approximating the solution. On the other side, given $|y_i - x_i\beta_0|$, we hope to choose the data points with larger $\|x_i\|$ values, since larger $\|x_i\|$ values probably cause the approximate solution be more efficient. From the computation view, calculating $\{\pi_i^0\}_{i=1}^n$ costs $O(nd)$, so the *gradient-based sampling* is much saving computational cost.

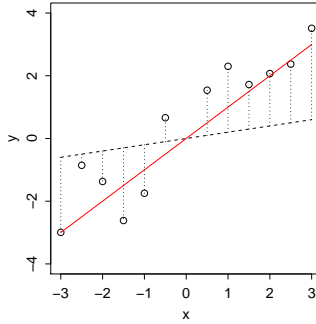

Figure 1: An illustration example. 12 data points are generated from $y_i = x_i + e_i$ where $x_i = (\pm 3, \pm 2.5, \pm 2, \pm 1.5, \pm 1, \pm 0.5)$ and $e_i \sim N(0, 0.5)$. The LS solution denoted by the red line $\hat{\beta} = \sum_{i=1}^{12} x_i y_i / \sum_{i=1}^{12} x_i^2$. The pilot estimate denoted by dashed line $\beta_0 = 0.5$.

**Choosing the pilot estimate $\boldsymbol{\beta}_0$.** In many applications, there may be a natural choice of pilot estimate $\boldsymbol{\beta}_0$, for instance, the fit from last time is a natural choice for this time. Another simple way is to use a pilot estimate $\boldsymbol{\beta}_0$ from an initial subsample of size $r_0$ obtained by uniform sampling. The extra computational cost is $O(r_0 d^2)$, which is assumed to be small since a choice $r_0 \le r$ will be good

enough. We empirically show the effect of small $r_0$ ($r_0 \leq r$) on the performance of the *gradient-based sampling* by simulations, and argue that one does not need to be careful when choosing $r_0$ to get a pilot estimate. (see Supplementary Material, Section S1)

**Poisson sampling v.s. sampling with replacement.** In this study, we do not choose *sampling with replacement* as did in previous studies, but apply *Poisson sampling* into this algorithm. *Poisson sampling* is executed in the following way: proceed down the list of elements and carry out one randomized experiment for each element, which results either in the election or in the nonselection of the element [16]. Thus, *Poisson sampling* can improve the efficiency in some context compared to sampling with replacement since it can avoid repeatedly drawing the same data points, especially when the sampling ratio increases, We empirically illustrates this advantage of Poisson sampling compared to sampling with replacement. (see Supplementary Material, Section S2)

**Independence on model assumption.** LS solution is well known to be statistically efficient under the linear regression model with homogeneous errors, but model misspecification is ubiquitous in real applications. On the other hand, LS solution is also an optimization problem without any linear model assumption from the algorithmic view. To numerically show the independence of the *gradient-based sampling* on model assumption, we do simulation studies and find that it is an efficient sampling method from the algorithmic perspective. (see Supplementary Material, Section S3)

Now as a summary we present the *gradient-based sampling* in Algorithm 1.

---

**Algorithm 1** *Gradient-based sampling* Algorithm

---

- **Pilot estimate $\beta_0$:**
  (1) Have a good guess as the pilot estimate $\boldsymbol{\beta}_0$, or use the initial estimate $\boldsymbol{\beta}_0$ from an initial subsample of size $r_0$ by uniform sampling as the pilot estimate.

- **Gradient-based sampling:**
  (2) Assign sampling probabilities $\{\pi_i \propto \|\boldsymbol{g}_i\|\}_{i=1}^n$ for all data points such that $\sum\limits_{i=1}^n \pi_i = 1$.
  (3) Generate independent $s_i \sim \text{Bernoulli}(1, p_i)$, where $p_i = r\pi_i$ and $r$ is the *expected* subsample size.
  (4) Get a subsample by selecting the element corresponding to $\{s_i = 1\}$, that is, if $s_i = 1$, the $i$th data is chosen, otherwise not.

- **Estimation:**
  (5) Solve the LS problem using the subsample using equation (3) then get the subsample estimator $\tilde{\boldsymbol{\beta}}$.

---

*Remarks on Algorithm 1.* (a) The subsample size $r^*$ from Poisson sampling is random in Algorithm 1. Since $r^*$ is multinomial distributed with expectation $E(r^*) = \sum_{i=1}^n p_i = r$ and variance $Var(r^*) = \sum_{i=1}^n p_i(1-p_i)$, the range of probable values of $r^*$ can be assessed by an interval. In practice we just need to set the *expected* subsample size $r$. (b) If $\pi_i$'s are so large that $p_i = r\pi_i > 1$ for some data points, we should take $p_i = 1$, i.e., $\pi_i = 1/r$ for them.

## 4  Error Bound Analysis of Sampling Algorithms

Our main theoretical result establishes the excess risk, i.e., an upper error bound of the subsample estimator $\tilde{\boldsymbol{\beta}}$ to approximate $\hat{\boldsymbol{\beta}}_n$ for an random sampling method. Given sampling probabilities $\{\pi_i\}_{i=1}^n$, the excess risk of the subsample estimator $\tilde{\boldsymbol{\beta}}$ with respect to $\hat{\boldsymbol{\beta}}_n$ is given in Theorem 1. (see Section S4 in Supplementary Material for the proof). By this general result, we provide an explanation why the *gradient-based sampling* algorithm is statistically efficient.

**Theorem 1** *Define $\sigma_\Sigma^2 = \frac{1}{n^2} \sum\limits_{i=1}^n \pi_i^{-1} \|\boldsymbol{x}_i\|^4$, $\sigma_b^2 = \frac{1}{n^2} \sum\limits_{i=1}^n \frac{1}{\pi_i} \|\boldsymbol{x}_i\|^2 e_i^2$ where $e_i = y_i - \boldsymbol{x}_i^T \hat{\boldsymbol{\beta}}_n$, and $R = \max\{\|\boldsymbol{x}_i\|^2\}_{i=1}^n$, if*

$$r > \frac{\sigma_\Sigma^2 \log d}{\delta^2 (2^{-1}\lambda_{min}(\boldsymbol{\Sigma}_n) - (3n\delta)^{-1} R \log d)^2}$$

*holds, the excess risk of $\tilde{\boldsymbol{\beta}}$ for approximating $\hat{\boldsymbol{\beta}}_n$ is bounded in probability $1 - \delta$ for $\delta > \frac{R \log d}{3n\lambda_{min}(\boldsymbol{\Sigma}_n)}$
as*

$$\|\tilde{\boldsymbol{\beta}} - \hat{\boldsymbol{\beta}}_n\| \leq Cr^{-1/2}, \tag{6}$$

*where $C = 2\lambda_{min}^{-1}(\boldsymbol{\Sigma}_n)\delta^{-1}\sigma_b$.*

Theorem 1 indicates that, $\|\tilde{\boldsymbol{\beta}} - \hat{\boldsymbol{\beta}}_n\|$ can be bounded by $Cr^{-1/2}$. From (6), the choice of sampling method has no effect on the decreasing rate of the bound, $r^{-1/2}$, but influences the constant $C$. Thus, a theoretical measure of efficiency for some sampling method is whether it can make the constant $C$ attain its minimum. In Corollary 1 (see Section S5 in Supplementary Material for the proof), we show that Algorithm 1 can approximately get this aim.

*Remarks on Theorem 1.* (a) Theorem 1 can be used to guide the choice of $r$ in practice so as to guarantee the desired accuracy of the solution with high probability. (b) The constants $\sigma_b$, $\lambda_{\min}(\boldsymbol{\Sigma}_n)$ and $\sigma_\Sigma$ can be estimated based on the subsample. (c) The risk of $\mathbf{X}\tilde{\boldsymbol{\beta}}$ to predict $\mathbf{X}\hat{\boldsymbol{\beta}}_n$ follows from equation (6) and get that $\|\mathbf{X}\tilde{\boldsymbol{\beta}} - \mathbf{X}\hat{\boldsymbol{\beta}}_n\|/n \leq Cr^{-1/2}\lambda_{\max}^{1/2}(\boldsymbol{\Sigma}_n)$. (d) Although Theorem 1 is established under Poisson sampling, we can easily extend the error bound to *sampling with replacement* by following the technical proofs in Supplementary Material, since each drawing in *sampling with replacement* is considered to be independent.

**Corollary 1** *If $\boldsymbol{\beta}_0 - \hat{\boldsymbol{\beta}}_n = o_p(1)$, then $C$ is approximately mimimized by Algorithm 1, that is,*

$$C(\pi_i^0) - \min_\pi C = o_p(1), \tag{7}$$

*where $C(\pi_i^0)$ denotes the value $C$ corresponding to our* gradient-based sampling.

The significance of Corollary 1 is to give an explanation why the *gradient-based sampling* is statistically efficient. The corollary establishes an improved performance guarantee on the use of the *gradient-based sampling*. It is improved in the sense that our *gradient-based sampling* can make the bound approximately attain its minimum as long as the condition is satisfied, while neither uniform sampling nor leverage-based sampling can get this aim. The condition that $\boldsymbol{\beta}_0 - \hat{\boldsymbol{\beta}}_n = o_p(1)$ provides a benchmark whether the pilot estimate $\boldsymbol{\beta}_0$ is a good guess of $\hat{\boldsymbol{\beta}}_n$. Note the condition is satisfied by the initial estimate $\boldsymbol{\beta}_0$ from an initial subsample of size $r_0$ by uniform sampling since $\boldsymbol{\beta}_0 - \hat{\boldsymbol{\beta}}_n = O_p(r_0^{-1/2})$.

## 5 Numerical Experiments

Detailed numerical experiments are conducted to compare the excess risk of $\tilde{\boldsymbol{\beta}}$ based on $L^2$ loss against the expected subsample size $r$ for different synthetic datasets and real data examples. In this section, we report several representative studies.

### 5.1 Performance of gradient-based sampling

The $n \times d$ design matrix $\mathbf{X}$ is generated with elements drawn independently from the mixture Gaussian distributions $\frac{1}{2}N(-\mu, \sigma_x^2) + \frac{1}{2}N(\mu, \theta_{mg}^2\sigma_x^2)$ below: (1) $\mu = 0$ and $\theta_{mg} = 1$, i.e., Gaussian distribution (referred as to GA data); (2) $\mu = 0$ and $\theta_{mg} = 2$, i.e.,the mixture between small and relatively large variances (referred as to MG1 data); (3) $\mu = 0$ and $\theta_{mg} = 5$, i.e., the mixture between small and highly large variances (referred as to MG2 data); (4) $\mu = 5$ and $\theta_{mg} = 1$, i.e., the mixture between two symmetric peaks (referred as to MG3 data). We also do simulations on $\mathbf{X}$ generated from multivariate mixture Gaussian distributions with AR(1) covariance matrix, but obtain the similar performance to the setting above, so we do not report them here. Given $\mathbf{X}$, we generate $\boldsymbol{y}$ from the model $\boldsymbol{y} = \mathbf{X}\boldsymbol{\beta} + \boldsymbol{\epsilon}$ where each element of $\boldsymbol{\beta}$ is drawn from normal distribution $N(0, 1)$ and then fixed, and $\boldsymbol{\epsilon} \sim N(\mathbf{0}, \sigma^2\mathbf{I}_n)$, where $\sigma = 10$. Note that we also consider the heteroscedasticity setting that $\boldsymbol{\epsilon}$ is from a mixture Gaussian, and get the similar results to the homoscedasticity setting. So we do not report them here. We set $d$ as 100, and $n$ as among 20K, 50K, 100K, 200K, 500K.

We calculate the full sample LS solution $\hat{\boldsymbol{\beta}}_n$ for each dataset, and repeatedly apply various sampling methods for $B = 1000$ times to get subsample estimates $\tilde{\boldsymbol{\beta}}_b$ for $b = 1, \dots, B$. We calculate the

empirical risk based on $L^2$ loss (MSE) as follows:

$$\text{MSE} = B^{-1} \sum_{b=1}^{B} \|\tilde{\boldsymbol{\beta}}_b - \hat{\boldsymbol{\beta}}_n\|^2.$$

Two sampling ratio $r/n$ values are considered: 0.01 and 0.05. We compare uniform sampling (UNIF), the leverage-based sampling (LEV) and the gradient-based sampling (GRAD) to these data sets. For GRAD, we set the $r_0 = r$ to getting the pilot estimate $\boldsymbol{\beta}_0$.

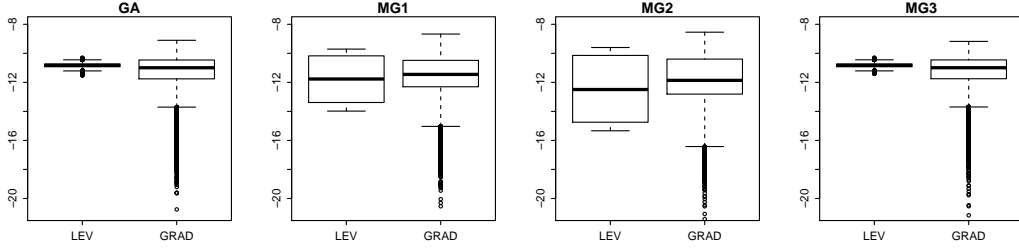

Figure 2: Boxplots of the logarithm of different sampling probabilities of $\mathbf{X}$ matrices with $n = 50K$. From left to right: GA, MG1, MG2 and MG3 data sets.

Figure 2 gives boxplots of the logarithm of sampling probabilities of LEV and GRAD, where taking the logarithm is to clearly show their distributions. We have some observations from the figure. (1) For all four datasets, GRAD has heavier tails than LEV, that is, GRAD lets sampling probabilities more disperse than LEV. (2) MG2 tends to have the most heterogeneous sampling probabilities, MG1 has less heterogeneous than MG2, whereas MG3 and GA have the most homogeneous sampling probabilities. This indicates that the mixture of large and small variances has effect on the distributions of sampling probabilities while the mixture of different peak locations has no effect.

We plot the logarithm of MSE values for GA, MG1, and MG2 in Figure 3, where taking the logarithm is to clearly show the relative values. We do not report the results for MG3, as there is little difference between MG3 and GA. There are several interesting results shown in Figure 3. (1) GRAD has better performance than others, and the advantage of GRAD becomes obvious as $r/n$ increases. (2) For GA, LEV is shown to have similar performance to UNIF, however GRAD has obviously better performance than UNIF. (3) When $r/n$ increases, the smaller $n$ is needed to make sure that GRAD outperforms others.

From the computation view, we compare the computational cost for UNIF, approximate LEV (ALEV) [5, 6] and GRAD in Table 1, since ALEV is shown to be computationally efficient to approximate LEV. From the table, UNIF is the most saving, and the time cost of GRAD is much less than that of ALEV. It indicates that GRAD is also an efficient method from the computational view, since its running time is $O(nd)$. Additionally, Table 2 summaries the computational complexity of several sampling methods for fast solving LS problems.

### 5.2 Real Data Examples

In this section, we compare the performance of various sampling algorithms on two UCI datasets: CASP ($n = 45730, d = 9$) and OnlineNewsPopularity (NEWS) ($n = 39644, d = 59$). At first, we plot boxplots of the logarithm of sampling probabilities of LEV and GRAD in Figure 4. From it, similar to synthetic datasets, we know that the sampling probabilities of GRAD looks more dispersed compared to those of LEV.

The MSE values are reported in Table 3. From it, we have two observations below. First, GRAD has smaller MSE values than others when $r$ is large. Second, as $r$ increases, the outperformance of Poisson sampling than sampling with replacement gets obvious for various methods. Similar observation is gotten in simulations (see Supplementary Material, Section S2).

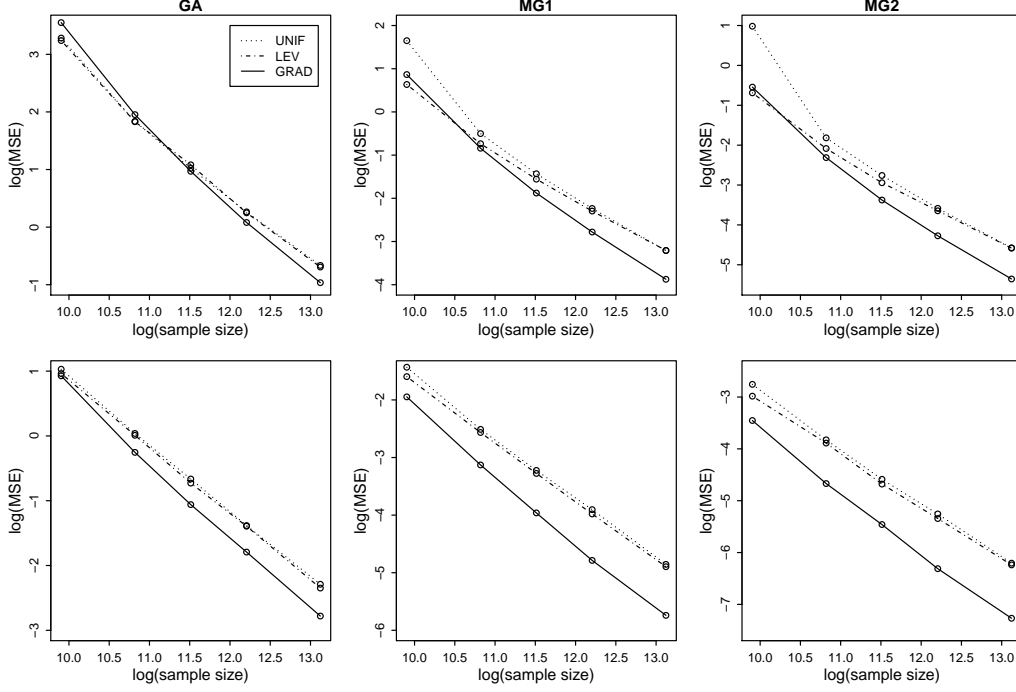

Figure 3: Empirical mean-squared error of $\tilde{\boldsymbol{\beta}}$ for approximating $\hat{\boldsymbol{\beta}}_n$. From top to bottom: upper panels are $r/n = 0.01$, and lower panels $r/n = 0.05$. From left to right: GA, MG1, and MG2 data, respectively.

Table 1: The cost time of obtaining $\tilde{\boldsymbol{\beta}}$ on various subsample sizes $r$ by UNIF, ALEV and GRAD for $n = 500K, 5M$, where () denotes the time of calculating full sample LS solution $\hat{\boldsymbol{\beta}}_n$. We perform the computation by R software in PC with 3 GHz intel i7 processor, 8 GB memory and OS X operation system.

| $n = 500K$ | | | | | | |
|---|---|---|---|---|---|---|
| | System Time (0.406) | | | User Time (7.982) | | |
| $r$ | 200 | 500 | 2000 | 200 | 500 | 2000 |
| UNIF | 0.000 | 0.002 | 0.003 | 0.010 | 0.018 | 0.050 |
| ALEV | 0.494 | 0.642 | 0.797 | 2.213 | 2.592 | 4.353 |
| GRAD | 0.099 | 0.105 | 0.114 | 0.338 | 0.390 | 0.412 |
| $n = 5M$ | | | | | | |
| | System Time (121.4) | | | User Time (129.88) | | |
| $r$ | 500 | 2000 | 10000 | 500 | 2000 | 10000 |
| UNIF | 0.057 | 0.115 | 0.159 | 2.81 | 5.94 | 14.28 |
| ALEV | 50.86 | 53.64 | 81.85 | 86.12 | 88.36 | 120.15 |
| GRAD | 5.836 | 6.107 | 6.479 | 28.85 | 30.06 | 37.51 |

# 6   Conclusion

In this paper we have proposed *gradient-based sampling* algorithm for approximating LS solution. This algorithm is not only statistically efficient but also computationally saving. Theoretically, we provide the error bound analysis, which supplies a justification for the algorithm and give a tradeoff between the subsample size and approximation efficiency. We also argue from empirical studies that: (1) since the *gradient-based sampling* algorithm is justified without linear model assumption, it works better than the *leverage-based sampling* under different model specifications; (2) Poisson sampling is much better than sampling with replacement when sampling ratio $r/n$ increases.

Table 2: The running time of obtaining $\tilde{\beta}$ by various sampling strategy. Stage D1 is computing the weights, D2 is computing the LS based on subsample, "overall" is the total running time.

| Stage | D1 | D2 | overall |
|---|---|---|---|
| Full | - | $O(\max\{nd^2, d^3\})$ | $O(\max\{nd^2, d^3\})$ |
| UNIF | - | $O(\max\{rd^2, d^3\})$ | $O(\max\{rd^2, d^3\})$ |
| LEV | $O(nd^2)$ | $O(\max\{rd^2, d^3\})$ | $O(\max\{nd^2, rd^2, d^3\})$ |
| ALEV | $O(nd\log n)$ | $O(\max\{rd^2, d^3\})$ | $O(\max\{nd\log n, rd^2, d^3\})$ |
| GRAD | $O(nd)$ | $O(\max\{rd^2, d^3\})$ | $O(\max\{nd, rd^2, d^3\})$ |

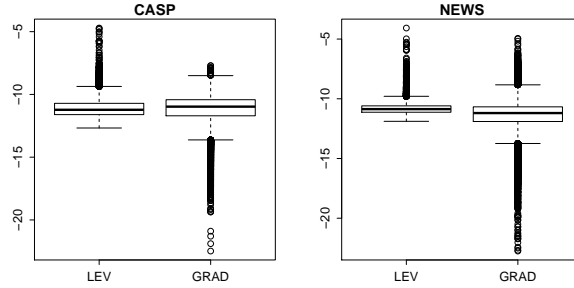

Figure 4: Boxplots of the logarithm of sampling probabilities for LEV and GRAD among datasets CASP and NEWS

Table 3: The MSE comparison among various methods for real datasets, where "*SR*" denotes sampling with replacement, and "*PS*" denotes Poisson sampling.

| | CASP $n = 45730, d = 9$ | | | | |
|---|---|---|---|---|---|
| $r$ | 45 | 180 | 450 | 1800 | 4500 |
| UNIF-*SR* | 2.998e-05 | 9.285e-06 | 4.411e-06 | 1.330e-06 | 4.574e-07 |
| UNIF-*PS* | 2.702e-05 | 9.669e-06 | 4.243e-06 | 1.369e-06 | 4.824e-07 |
| LEV-*SR* | **1.962e-05** | **4.379e-06** | 1.950e-06 | 4.594e-07 | 2.050e-07 |
| LEV-*PS* | 2.118e-05 | 5.240e-06 | 1.689e-06 | 4.685e-07 | 1.694e-07 |
| GRAD-*SR* | 2.069e-05 | 5.711e-06 | 1.861e-06 | 4.322e-07 | 1.567e-07 |
| GRAD-*PS* | 2.411e-05 | 5.138e-06 | **1.678e-06** | **3.687e-07** | **1.179e-07** |
| | NEWS $n = 39644, d = 59$ | | | | |
| $r$ | 300 | 600 | 1200 | 2400 | 4800 |
| UNIF-*SR* | 22.050 | 14.832 | 10.790 | 7.110 | 4.722 |
| UNIF-*PS* | 27.215 | 19.607 | 15.258 | 9.504 | 4.378 |
| LEV-*SR* | 22.487 | 11.047 | 5.519 | 2.641 | 1.392 |
| LEV-*PS* | 21.971 | 9.419 | 4.072 | 2.101 | 0.882 |
| GRAD-*SR* | 10.997 | 5.508 | 3.074 | 1.505 | 0.752 |
| GRAD-*PS* | **9.729** | **5.252** | **2.403** | **1.029** | **0.399** |

There is an interesting problem to address in the further study. Although the *gradient-based sampling* is proposed to approximate LS solution in this paper, we believe that this sampling method can apply into other optimization problems for large-scale data analysis, since gradient is considered to be the steepest way to attain the (local) optima. Thus, applying this idea to other optimization problems is an interesting study.

## Acknowledgments

This research was supported by National Natural Science Foundation of China grants 11301514 and 71532013. We thank Xiuyuan Cheng for comments in a preliminary version.

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
