[Supplementary Material · supplementary.pdf]

# Supplementary Material of "Gradient-based Sampling: An Adaptive Importance Sampling for Least-squares"

## S.1 The influence of the pilot estimate

In *gradient-based sampling*, we need to get the pilot estimate $\boldsymbol{\beta}_0$ by uniformly sampling a subsample of size $r_0$. Now we investigate the effect of $r_0$ by plotting the relative MSE for $r_0 = \{0.1r, 0.2r, \cdots, 0.9r\}$ with respect to that for $r_0 = r$ on GA, MG1 and MG2 datasets in Figure 1. We observe that when $r_0$ is larger than $0.5r$, MSE values of $\tilde{\boldsymbol{\beta}}$ go flat. Thus, we argue that choosing the initial size of $r_0$ to get a pilot estimate may not be careful.

Figure 1: The relative MSE values of $\tilde{\boldsymbol{\beta}}$ with uniformly sampling $r_0 = \{0.1r, 0.2r, \cdots, 0.9r\}$ data points for the pilot estimate $\boldsymbol{\beta}_0$ with respect to that with $r_0 = r$. From left to right: GA, MG1 and MG2 datasets

## S.2 The advantage of poisson sampling

Now we empirically compare *poisson sampling (PS)* with *sampling with replacement (SR)*. We compare risk performance between them for different $r/n$ values: 0.01 and 0.05. We report the results in Table 1, where we do not report the performance of UNIF and LEV due to the similarity shared with GRAD. From Table 1, there is little difference between *PS* and *SR* for $r/n = 0.01$, however

Table 1: Ratios of MSE values by *PS* and these by *SR*.

| n | 20K | 50K | 100K | 200K | 500K |
|---|---|---|---|---|---|
| $s/n = 0.01$ | | | | | |
| GA | 1.061 | 0.945 | 0.986 | 0.946 | 1.019 |
| MG1 | 0.968 | 0.945 | 0.927 | 0.973 | 0.997 |
| MG2 | 1.020 | 0.980 | 0.969 | 0.989 | 1.004 |
| $s/n = 0.05$ | | | | | |
| GA | 0.920 | 0.946 | 0.937 | 0.975 | 0.970 |
| MG1 | 0.860 | 0.921 | 0.909 | 0.853 | 0.925 |
| MG2 | 0.885 | 0.843 | 0.895 | 0.824 | 0.835 |

*PS* becomes better than *SR* for $r/n = 0.05$. This observation indicates that *PS* outperforms *SR* when the sampling ratio $r/n$ increases.

## S.3   The Robustness to Model Specification

The *gradient-based sampling* algorithm does not reply on the model assumption. We empirically investigate the effect of the model specification on various sampling methods. Three kinds of model specification are considered here, i.e., models generating data are as follows:

(I) heteroscedasticity,

$$\boldsymbol{y} = \sum_{k=1}^{10} \boldsymbol{x}^{(k)}\beta_k + \boldsymbol{\varepsilon}^* \quad \text{with} \quad \boldsymbol{\varepsilon}^* = \rho_1 \boldsymbol{x}^{(11)} + \boldsymbol{\varepsilon},$$

where $\boldsymbol{x}^{(11)}$ is ignored in LS computation, and $\rho_1$ denotes the seriousness degree of "model wrong" and is set as among $\{0, 0.1, 0.2, 0.5, 1, 2, 5, 10\}$;

(II) model error dependence,

$$\boldsymbol{y} = \sum_{k=1}^{10} \boldsymbol{x}^{(k)}\beta_k + \boldsymbol{\varepsilon}, \quad \text{with} \quad \varepsilon_i = N\left(\rho_2 \varepsilon_{i-1}, (1 - \rho_2^2)\sigma^2\right),$$

where $\varepsilon_0 = 0$, and $\rho_2$ denotes the dependence degree among model errors and is set as among $\{0, 0.2, 0.4, 0.5, 0.6, 0.7, 0.8, 0.9\}$;

(III) correlation between error and predictor,

$$\boldsymbol{y} = \sum_{k=1}^{10} \boldsymbol{x}^{(k)}\beta_k + \boldsymbol{\varepsilon} \quad \text{with} \quad \varepsilon_i = \left(1 + \rho_3 x_i^{(1)}\right) N(0, \sigma^2),$$

Table 2: The performance of $\tilde{\boldsymbol{\beta}}$ for approximating $\hat{\boldsymbol{\beta}}_n$ under three kinds of model specification for MG1 dataset.

| Type I: heteroscedasticity | | | | | |
|---|---|---|---|---|---|
| $\rho_1$ | 0 | 0.2 | 0.5 | 1 | 2 |
| UNIF | 0.027 | 0.031 | 0.054 | 0.131 | 0.466 |
| LEV | 0.026 | 0.029 | 0.038 | 0.078 | 0.227 |
| GRAD | **0.013** | **0.016** | **0.026** | **0.060** | **0.199** |
| Type II: model error dependence | | | | | |
| $\rho_2$ | 0 | 0.2 | 0.5 | 0.7 | 0.9 |
| UNIF | 0.027 | 0.027 | 0.028 | 0.028 | 0.027 |
| LEV | 0.026 | 0.025 | 0.027 | 0.025 | 0.026 |
| GRAD | **0.013** | **0.0143** | **0.013** | **0.014** | **0.013** |
| Type III: correlation between error and predictor | | | | | |
| $\rho_3$ | 0 | 0.2 | 0.5 | 1 | 2 |
| UNIF | 0.027 | 0.545 | 3.181 | 12.74 | 51.07 |
| LEV | 0.026 | 0.235 | 1.344 | 5.271 | 20.80 |
| GRAD | **0.013** | **0.157** | **0.908** | **3.724** | **14.67** |

where $\rho_3$ denotes the correlation between model error and the predictor $\boldsymbol{x}^{(1)}$ and is set as among $\{0, 0.1, 0.2, 0.3, 0.5, 0.8, 1, 2\}$.

We report the results on MG1 dataset for $n = 50K$ and $r = 200$ in Table 2 but do not report the results on other data sets because of the similarity. From Table 2, Firstly, most importantly, GRAD still works better than UNIF and LEV. Secondly, Types I and III can bring serious effect, especially Type III causes the most serious effect, while Type II seems have little effect on efficiency of sampling methods. Thus, these observations command that GRAD is a nice choice from the model robustness viewpoint.

## S.4 Technical Results

### A Lemma for proving Theorem 1

To analyze the risk, our key point is to apply Matrix Bernstein expectation bound (Theorem 6.1 in [1]) into matrix Bernoulli series. The lemma below present the expectation bound for matrix Bernoulli series.

**Lemma 1.** *Consider a finite sequence* $\{\mathbf{A}_i = \boldsymbol{x}_i\boldsymbol{x}_i^T\}$ *of Hermitian matrices, where* $\boldsymbol{x}_i$ *is* $d \times 1$ *vector. Let* $\{\gamma_i\}$, *with mean* $p_i$ *respectively, be a finite sequence of independent Bernoulli variables. Let* $\max\{\|\boldsymbol{x}_i\|^2\}_{i=1}^n = R$ *and*

$$\sigma_\Sigma^2 = \frac{1}{n^2}\sum_{i=1}^n \pi_i^{-1}\|\boldsymbol{x}_i\|^4.$$

*Define matrix Bernoulli series* $\boldsymbol{\Delta} = n^{-1}\sum_i(\gamma_i/p_i - 1)\mathbf{A}_i$. *We have,*

$$E\lambda_{max}(\boldsymbol{\Delta}) \leq r^{-1/2}\sigma_\Sigma\sqrt{\log d} + \frac{R}{3n}\log d.$$

Since the sequence $\{n^{-1}(\gamma_i/p_i - 1)\mathbf{A}_i\}_{i=1}^n$ is independent random Hermitian matrices with $E[n^{-1}(\gamma_i/p_i - 1)\mathbf{A}_i] = 0$, $\lambda_{\max}(n^{-1}(\gamma_i/p_i - 1)\mathbf{A}_i) \leq \lambda_{\max}(n^{-1}\mathbf{A}_i) = R/n$, and

$$\lambda_{\max}(E\boldsymbol{\Delta}^2) = \frac{1}{n^2}\sum_{i=1}^n(p_i^{-1} - 1)\lambda_{\max}(\mathbf{A}_i^2)$$

$$\leq \frac{r^{-1}}{n^2}\sum_{i=1}^n \pi_i^{-1}\|\boldsymbol{x}_i\|^4,$$

applying the matrix Berstein inequality of Theorem 6.1 in ([1]) to obtain that

$$E\lambda_{max}(\boldsymbol{\Delta}) \leq r^{-1/2}\sigma_\Sigma\sqrt{\log d} + \frac{R}{3n}\log d.$$

## B    Proof of Theorem 1

We have that

$$\|\tilde{\boldsymbol{\beta}} - \hat{\boldsymbol{\beta}}\| = \|\boldsymbol{\Sigma}_s^{-1}\boldsymbol{b}_s - \boldsymbol{\Sigma}_s^{-1}\boldsymbol{\Sigma}_s\hat{\boldsymbol{\beta}}_n\|$$

$$\leq \lambda_{\max}(\boldsymbol{\Sigma}_s^{-1})\|\boldsymbol{b}_s - \boldsymbol{\Sigma}_s\hat{\boldsymbol{\beta}}_n\|. \tag{A.1}$$

Note that $\lambda_{\max}(\boldsymbol{\Sigma}_s^{-1}) - \lambda_{\max}(\boldsymbol{\Sigma}_n^{-1}) \leq \lambda_{\max}(\boldsymbol{\Sigma}_s^{-1} - \boldsymbol{\Sigma}_n^{-1}) \leq \lambda_{\max}(\boldsymbol{\Sigma}_s^{-1})\lambda_{\max}(\boldsymbol{\Sigma}_n^{-1})\lambda_{\max}(\boldsymbol{\Sigma}_s - \boldsymbol{\Sigma}_n)$, so if the event

$$\mathcal{E}_1 := \{\lambda_{\max}(\boldsymbol{\Sigma}_s - \boldsymbol{\Sigma}_n) < 2^{-1}\lambda_{\min}(\boldsymbol{\Sigma}_n)\} \tag{A.2}$$

holds, then we have that

$$\lambda_{\max}(\boldsymbol{\Sigma}_s^{-1}) \leq [\lambda_{\min}(\boldsymbol{\Sigma}_n) - \lambda_{\max}(\boldsymbol{\Sigma}_s - \boldsymbol{\Sigma}_n)]^{-1},$$

and combining (A.1),

$$\|\tilde{\boldsymbol{\beta}} - \hat{\boldsymbol{\beta}}\| \leq \frac{\|\boldsymbol{b}_s - \boldsymbol{\Sigma}_s\hat{\boldsymbol{\beta}}_n\|}{\lambda_{\min}(\boldsymbol{\Sigma}_n) - \lambda_{\max}(\boldsymbol{\Sigma}_s - \boldsymbol{\Sigma}_n)} < [\lambda_{\min}^{-1}(\boldsymbol{\Sigma}_n) + 2\lambda_{\min}^{-2}(\boldsymbol{\Sigma}_n)\lambda_{\max}(\boldsymbol{\Sigma}_s - \boldsymbol{\Sigma}_n)]\|\boldsymbol{b}_s - \boldsymbol{\Sigma}_s\hat{\boldsymbol{\beta}}_n\|, \tag{A.3}$$

where the 2nd inequality is from the fact that $\frac{1}{1-x} < 1 + 2x$ for any $0 < x < 1/2$ and the condition that the event $\mathcal{E}_1$ holds. For any $\delta > 0$, define

$$\mathcal{E}_2 := \{\|\boldsymbol{b}_s - \boldsymbol{\Sigma}_s\hat{\boldsymbol{\beta}}_n\| \geq \frac{\sigma_b}{r^{1/2}\delta}\}.$$

$$\mathcal{E}_3 := \{\lambda_{\max}(\boldsymbol{\Sigma}_s - \boldsymbol{\Sigma}_n) \geq \frac{\sigma_\Sigma\sqrt{\log d}}{r^{1/2}\delta} + \frac{R\log d}{3n\delta}\}.$$

Since

$$E\|\boldsymbol{b}_s - \boldsymbol{\Sigma}_s\hat{\boldsymbol{\beta}}_n\|^2$$

$$=E\left[\frac{1}{n^2}\sum_{i=1}^{n}\left(\frac{I_i}{p_i} - 1\right)\boldsymbol{x}_i^T e_i \sum_{i=1}^{n}\left(\frac{I_i}{p_i} - 1\right)\boldsymbol{x}_i e_i\right]$$

$$=\frac{1}{n^2}\sum_{i=1}^{n}\left(\frac{1}{p_i} - 1\right)\boldsymbol{x}_i^T \boldsymbol{x}_i e_i^2 < \frac{1}{r}\sigma_b^2,$$

by Markov's inequality we have that,

$$Pr(\mathcal{E}_2) \leq \delta. \tag{A.4}$$

Lemma 1 shows that

$$Pr(\mathcal{E}_3) \leq (\frac{\sigma_\Sigma\sqrt{\log d}}{r^{1/2}\delta} + \frac{R\log d}{3n\delta})^{-1}[\lambda_{max}(\boldsymbol{\Sigma}_s - \boldsymbol{\Sigma}_n)]$$

$$= \delta. \tag{A.5}$$

For (A.2), we have that $\mathcal{E}_1 \subseteq \mathcal{E}_3$ if

$$r > \frac{\sigma_\Sigma^2 \log d}{\delta^2(2^{-1}\lambda_{\min}(\boldsymbol{\Sigma}_n) - (3n\delta)^{-1}R\log d)^2}, \tag{A.6}$$

$$\delta > \frac{2R\log d}{3n\lambda_{\min}(\boldsymbol{\Sigma}_n)} \tag{A.7}$$

holds. Thus, combing (A.3), (A.4), (A.5), (A.6) and (A.7), we get

$$Pr\left\{\|\tilde{\boldsymbol{\beta}} - \boldsymbol{\beta}\| \geq C_1 r^{-1/2} + C_2 r^{-1}\right\} \leq Pr\left\{\mathcal{E}_2 \bigcap \mathcal{E}_3\right\} \leq \delta,$$

where $C_1 = \lambda_{\min}^{-1}(\boldsymbol{\Sigma}_n)\delta^{-1}\sigma_b$ and $C_2 = 2\lambda_{\min}^{-2}(\boldsymbol{\Sigma}_n)\delta^{-2}\sigma_\Sigma\sigma_b$. From (A.6), $C_1 r^{-1/2} + C_2 r^{-1} < 2C_1 r^{-1/2}$. Thus Theorem 1 is proved.

## C  Proof of Corollary 1

Let

$$\pi_i^e = \|e_i\boldsymbol{x}_i\|/\sum_{j=1}^{n}\|e_j\boldsymbol{x}_j\|. \tag{A.8}$$

$\sigma_b^2$ is minimized at $\{\pi_i^e\}_{i=1}^n$ by Cauchy-Schwarz inequality, and the minimum of $\sigma_b^2$:

$$\sigma_b^2(\pi_i^e) = (\frac{1}{n} \sum_{i=1}^n \|e_i \boldsymbol{x}_i\|)^2. \tag{A.9}$$

On the other hand, for the sampling probabilities $\pi_i^0$ of the gradient-based sampling,

$$\sigma_b^2(\pi_i^0) = (\frac{1}{n} \sum_{i=1}^n \|e_i \boldsymbol{x}_i\|)(\frac{1}{n} \sum_{i=1}^n \|\boldsymbol{x}_i\| e_i^2 / |\tilde{e}_i|), \tag{A.10}$$

where $\tilde{e}_i = y_i - \boldsymbol{x}_i^T \boldsymbol{\beta}_0$. From (A.9) and (A.10), we have that, if $\boldsymbol{\beta}_0 - \hat{\boldsymbol{\beta}}_n = o_p(1)$, then

$$\sigma_b^2(\pi_i^0) - \sigma_b^2(\pi_i^e) = o_p(1). \tag{A.11}$$

From (A.11) and the notation of $C$, Corollary 1 is proved.

# References

[1] J.A. Tropp. User-friendly tools for random matrices: An introduction. In *Advances in Neural Information Processing Systems*, 2012.