[Reviews · NeurIPS 2016]

Reviewer 1

Summary

The authors propose a gradient-based sub-sampling method for least-squares optimization. The method consists of drawing an initial subset of samples using a uniform distribution, though I assume any distribution could be used here, and using this set to obtain an initial estimate of the predictor coefficients. The gradients of this predictor wrt to the samples in the original set are then used to define a probability distribution with which a new subset is drawn and used for training.

Qualitative Assessment

The authors present both a theoretical and an empirical analysis of the proposed method. The empirical analysis is rather limited, as often happens with NIPS submissions. It would seem that the proposed method performs slightly better for large r on the CASP dataset while the improvement over the baseline is more evident on the NEWS dataset. The experiments presented in the supplementary material would seem to establish empirically the proposed method's sensitivity to the size of the initially drawn subset. This is not surprising both intuitively and in view of Corollary 1. Regarding the theoretical analysis, the main contribution would seem to be Corollary 1. Theorem 1 is independent of the sub-sampling distribution and is similar in nature to bounds derived elsewhere in the literature (e.g. http://www.drineas.org/Papers/Drineas_SODA_06.pdf). Regarding Corollary 1, it would be helpful if the authors could elaborate further. One issue that arises is the claim, line 146, that the sub-sampling method has no influence on the value r. However, line 142, r would seem to depend on \sigma_\Sigma which in turn depends on the probabilities \pi_i. Perhaps the authors could clarify this point. Another issue is the claim, line 164, that neither uniform nor leverage sampling can "get this aim", no proof is presented for this claim nor any citation given. Some clarification would be appreciated. Finally, perhaps due to unfamiliar notation, it is not clear to me what \beta - \beta_n = o_P (1) means exactly, I believe I understand the general concept, but an alternative, precise, mathematical definition would help. In particular, it is not clear to me how one moves from A.9, A.10 to A.11. Perhaps the proof could be fleshed out here. On a final note, gradient based sampling, is not in itself a novel idea. For example there exists prior work analyzing such approaches in the context of SGD (e.g. https://arxiv.org/pdf/1401.2753v2.pdf). As such the contribution of the work would seem to be the specific method of using gradient-based sampling for least squares and in particular the theoretical analysis of the approach.

Confidence in this Review

2-Confident (read it all; understood it all reasonably well)


Reviewer 2

Summary

In this work, the authors propose a novel technique in order to select a subset of the available data, reducing the total number of used data, performing the least-square (LS) solution in a faster way. The idea is to associate a normalized weight to each sample, in order to determine the "importance" of this sample in the LS solution. How to construct these weights is the key component of the proposed random sampling algorithm. The authors propose a gradient-based solution.

Qualitative Assessment

I enjoy the reading of this paper. The topic is very interesting and the work seems technically correct.

Confidence in this Review

1-Less confident (might not have understood significant parts)


Reviewer 3

Summary

The authors propose a sub-sampling algorithm for least squares regression, where the weights depend on the magnitude of the gradient. They argue that by using these weights, rather than uniform weights or weights based on leverage, the resulting regression estimates are more accurate. They bound the error of the resulting regression coefficients and they show that their method is optimal, in the sense that it minimizes a constant. They illustrate the performance of the method using a simulated example and two real datasets.

Qualitative Assessment

I don't think that the paper provides convincing arguments as to why using the gradient to determine the sub-sampling probabilities is a good idea. In particular, I worry that selecting mostly points characterised by big residuals will lead to un-robust regression estimates. In addition, are the weights invariant to the scale of the covariates? The quadratic form used to calculate the leverage is invariant, but I don't think that the gradient is. Hence, the results might depend on the scale used. Looking at Example 1: notice that the errors are homoscedastic here. I guess that under heteroscedasticity gradient-based sampling would do quite badly. Under heteroscedasticity one typically wants to down-weight observations where the variance is large. On the contrary, gradient-based sampling will give large weights to these points, because they have big residuals. This issue should be particularly problematic when the number of sub-samples (r) is small. Hence, before using the proposed strategy one has to be quite sure that the errors are homoscedastic. The title might be a bit misleading at the moment: I don't see in which sense the proposed approach is adaptive and I also don't see where the importance sampling part comes from (there seems to be no notion of an importance and a target distribution here). I would say that the paper focuses simply on gradient-based sub-sampling. In Theorem 1: I find a bit puzzling that if we plug r = n in formula (6) we do not get zero that the error is zero, as we should. In Section 5.1. How large is d here? The number of covariates is not reported. MINOR POINTS: L 15: "has obvious advantage" use either "has obvious advantages" or "has an obvious advantage". L18: "Modern data analysis always meets enormous data sets in recent years." maybe "meets" is not the correct word here. L22: "uniform sampling is simple but trivial way" maybe better "uniform sampling is a simple but trivial way". L28-29: "Another shortcoming is that it needs to cost much time" better "Another shortcoming is that it costs much time". L46: "we establishes" L64: "with d-dimensional predictors" here there are d predictors, but they are not d-dimensional L114: "and argue that choosing the initial size of r 0 to get a pilot estimate may not be careful" do you mean that one does not need to be careful when choosing r0? L116: Poisson should be have capital P. L128: ", we do simulation studies and command that it is an efficient sampling method from the algorithmic perspective." maybe you mean "find" rather than "command". L211: "we compare the performance of various sampling on two" add "algorithms" after "sampling". Table 3: what are the TS-PS entries in the CASP table? L223: "which supply a rationality for the algorithm" maybe "justification" rather than "rationality". Also, "supplies" rather than "supply". L225: "since the gradient-based sampling algorithm is implied without linear model assumption" maybe "justified" rather than "implied" L231: "since gradient is considered a steepest way to go to the (local) optima." This sentence needs to be re-written. L231: "optimzation". L231: "Thus, applying this idea into other" use "to" rather than "into".

Confidence in this Review

2-Confident (read it all; understood it all reasonably well)


Reviewer 4

Summary

The paper proposes a new adaptive subsampling procedure for solving large least-squares problems. It uses the gradient of the objective with respect to an initial rough estimate of the regression coefficients, and thus depends on both the predictors and responses. The subsampling procedure is more efficient than previous procedures depending on the statistical leverage scores. The excess risk (with respect to the least squares estimate on the entire data set) of the algorithm's estimate is bounded with respect to the least squares estimate on the entire data set. Empirical results show that the proposed algorithm achieves a lower excess risk on average than simple uniform sampling or leverage-based sampling.

Qualitative Assessment

The proposed subsampling procedure is simple, interesting, and novel. It and its theoretical results are complete and presented clearly. I believe that in certain situations, the proposed procedure could be a significant improvement to current algorithms. I like the idea of using Poisson sampling, and I like that results (computation time and estimation error) on both simulated and real data were presented. I have a few questions/concerns: 1. The sample size may be, in bad cases, on the same order of magnitude as the data set size. This would not significantly reduce computation cost, so it would be helpful to provide some concentration bounds on the sample size. 2. In section 5.1, why we care about the distribution of sampling probabilities? 3. In the real data examples part of Section 5, it would be helpful to have running times of the various algorithms as well. For example, does Poisson sampling increase or decrease the computation time? Overall, the authors clearly communicate their ideas; however, please have the manuscript proof-read for English grammatical and style issues. For example, in the introduction, please define the 'gradient value'. It is somewhat confusing because it is not clear what gradient it is referring to. On an aesthetic note, some figures seem too small.

Confidence in this Review

2-Confident (read it all; understood it all reasonably well)


Reviewer 5

Summary

This paper provides a method to approximate least-square solution based on the contribution of the gradient of each instance. The instance which has larger gradient should make more contribution to decreasing to the minimum and have a larger weight. Compared with LEV, GRAD not only saves computational cost but also has better numerical performance in some empirical experiments. GRAD also has much better performance than UNIF.

Qualitative Assessment

The idea of including the output information on least-square problems is new. The author has provides theoretical and numerical justification on it. But I also have some questions when I read this paper. First, p_i=r \pi_i, is it possible that p_i bigger than one. In this case, we actually let p_i=1, then the expected sample size is not r. How to deal with this case? Second, it looks like ALEV has a comparable time complexity with GRAD. In numerical experiments in Figure 3, this paper only compare performance with LEV. Could you also compare numerical performance with ALEV? Third, the comparison with another type of popular method, stochastic gradient descent method, is not clear. Could the author provide some comparison with stochastic gradient descent method. From my perspective, there are some mirror mistakes when I read the papers. In line 73, "minimize" a weight loss function instead of "maximize". Second, on equation (4), the calculation of the gradient is not right although it will not affect the importance weights.

Confidence in this Review

2-Confident (read it all; understood it all reasonably well)


Reviewer 6

Summary

This paper proposes a method for solving the least squares problem via non-uniform sub-sampling based algorithm. The main contribution is the sampling technique which relies on the magnitudes of the gradients to assign sampling probabilities. The authors establish a theoretical bound on the excess risk, and empirically validate the performance of their algorithm on synthetic and real datasets.

Qualitative Assessment

The main concern is that there are lots of typos and grammatical issues in the paper that it is extremely hard to follow. It should be proof-read for English grammatical issues. The authors mention about "statistical efficiency" of the proposed method. I believe this is different than the term "efficiency" in statistics. Can you clarify this?

Confidence in this Review

3-Expert (read the paper in detail, know the area, quite certain of my opinion)